# Epigenetic Regulation of NRF2/KEAP1 by Phytochemicals

**DOI:** 10.3390/antiox9090865

**Published:** 2020-09-14

**Authors:** Shamee Bhattacharjee, Roderick H. Dashwood

**Affiliations:** 1Department Zoology, West Bengal State University, Kolkata 700126, India; shamee.zoology@wbsu.ac.in; 2Center for Epigenetics & Disease Prevention, Texas A&M Health Science Center, Department Translational Medical Sciences, Texas A&M College of Medicine, Houston, TX 77030, USA

**Keywords:** antioxidant response element, cancer interception, DNA methylation, histone acetylation, histone methylation, lncRNA, miRNA

## Abstract

Epigenetics has provided a new dimension to our understanding of nuclear factor erythroid 2–related factor 2/Kelch-like ECH-associated protein 1 (human NRF2/KEAP1 and murine Nrf2/Keap1) signaling. Unlike the genetic changes affecting DNA sequence, the reversible nature of epigenetic alterations provides an attractive avenue for cancer interception. Thus, targeting epigenetic mechanisms in the corresponding signaling networks represents an enticing strategy for therapeutic intervention with dietary phytochemicals acting at transcriptional, post-transcriptional, and post-translational levels. This regulation involves the interplay of histone modifications and DNA methylation states in the human *NFE2L2/KEAP1* and murine *Nfe2l2/Keap1* genes, acetylation of lysine residues in NRF2 and Nrf2, interaction with bromodomain and extraterminal domain (BET) acetyl “reader” proteins, and non-coding RNAs such as microRNA (miRNA) and long non-coding RNA (lncRNA). Phytochemicals documented to modulate NRF2 signaling act by reversing hypermethylated states in the CpG islands of *NFE2L2* or *Nfe2l2*, via the inhibition of DNA methyltransferases (DNMTs) and histone deacetylases (HDACs), through the induction of ten-eleven translocation (TET) enzymes, or by inducing miRNA to target the 3′-UTR of the corresponding mRNA transcripts. To date, fewer than twenty phytochemicals have been reported as NRF2 epigenetic modifiers, including curcumin, sulforaphane, resveratrol, reserpine, and ursolic acid. This opens avenues for exploring additional dietary phytochemicals that regulate the human epigenome, and the potential for novel strategies to target NRF2 signaling with a view to beneficial interception of cancer and other chronic diseases.

## 1. Introduction

The NRF2 signaling axis has received widespread attention from the research community due to its critical role in responding to xenobiotic and electrophilic stress [1]. Binding of NRF2 to antioxidant response element (ARE) sequences in gene promoters activates antioxidant and xenobiotic detoxifying enzymes [2]. Activation of NRF2 by various dietary phytochemicals provides a promising strategy to prevent cancer, and the protective role of Nrf2/NRF2 activators has been verified in preclinical models and in human clinical trials [3,4].

However, inactivating mutations in *KEAP1* that lead to constitutive NRF2 activation [5,6] can provide a growth advantage in some cancer cells. The yin/yang aspect of NRF2 signaling was first elaborated by Wang et al., reporting that knockdown of *NRF2* using siRNA, or stable overexpression of *KEAP1*, sensitized human cancer cells to chemotherapy [7]. There is growing evidence linking elevated NRF2 expression with chemoresistance and poor prognosis in various cancer types [8,9].

In view of this functional duality, NRF2 has been discussed both as a “friend or foe” or a “double-edged sword” in cancer etiology [10,11]. However, the context and timing of NRF2 activation plays an important role in determining beneficial outcomes and highlights the critical need for a thorough mechanistic understanding of NRF2 signaling [12].

Regulation of NRF2 signaling occurs at the level of transcription, post-transcription, and protein stability. Although genetic alterations initially were reported in *NFE2L2*/*KEAP1*, more recently, epigenetic mechanisms have added a new dimension and an element of fine-tuning to the NRF2 signaling axis. Thus, the present review aims to provide an update on the various phytochemicals that regulate NRF2 via the “epigenetic trinity” of DNA methylation, histone modifications, and non-coding RNAs.

## 2. Multilayer Regulation of NRF2 Signaling

### 2.1. Transcriptional Regulation

A wide array of stimuli can activate NRF2 signaling, including oxidative, inflammatory, and metabolic stressors [13]. The *NFE2L2* and *Nfe2l2* genes are regulated at the transcriptional level by multiple transcription factors [14]. For example, the Aryl hydrocarbon receptor/Aryl hydrocarbon receptor nuclear translocator (AhR/Arnt) heterodimer can interact with a xenobiotic response element in the *NFE2L2* promoter, leading to transactivation [15]. Also, in response to inflammatory stimuli, *Nfe2l2* can be activated by nuclear factor kappa B (NF-κB) [16], whereas a c-Jun binding site in *Nfe2l2* was implicated in oncogenic Nrf2 activation via K-ras, B-Raf, and c-Myc [17]. Notch signaling directly activated Nrf2 by recruiting Notch Intracellular Domain-recombination signal binding protein for the immunoglobulin kappa J region complex to a conserved site in the promoter of *Nfe2l2,* which promoted cytoprotective outcomes during liver development and hepatic stress responses [18]. Interestingly, Nrf2 also can regulate its own transcription by binding to ARE-like sequences in the *Nfe2l2* promoter [19].

### 2.2. Post-Transcriptional Regulation

Modifications of the *NFE2L2* mRNA transcript also play essential roles in the activation and correct functioning of NRF2. Post-transcriptional processing includes regulation by microRNAs (miRNAs), long non-coding RNAs (lncRNAs), adenosine methylation, and alternative splicing of the *NFE2L2* transcript. Several miRNAs downregulate or upregulate NRF2 protein expression and activity by directly targeting 3′-UTR sequences of *NFE2L2* or *KEAP1* mRNA, respectively. Evidence also has accrued linking lncRNAs to NRF2 regulation, as discussed in Section 3.4.

Alternative splicing of *NFE2L2* is another key mechanism regulating NRF2 activity. Some tumors express transcript variants of *NFE2L2* that lack exons coding for the KEAP1 interacting domain, resulting in hyperactivation of the NRF2 pathway [20]. A recent study reported regulation of *Nfe2l2* mRNA nuclear export and stabilization by two mRNA binding proteins, HuR and AUF1, targeting the 3′-UTR of the nascent transcript [21].

### 2.3. Regulation of NRF2 Protein Stability

Protein levels of NRF2 often are tightly regulated by proteasomal degradation complexes, in particular via the Cullin 3/RING-box protein 1 (Cul3/Rbx1)/Keap1 complex. KEAP1 acts as a linker protein between NRF2 and the Cul3/Rbx1-based ubiquitin ligase and causes continuous degradation under basal conditions, resulting in low constitutive NRF2 levels under physiological conditions. Regulation of NRF2 protein stability also is mediated by KEAP1-independent proteasomal degradation mechanisms, such as through the S-phase kinase-associated protein 1/Cullin/F-box (SCF)-β-transducin repeats-containing proteins (β-TrCP) complex, or HMG-CoA reductase degradation protein 1 (HRD1) [14]. Degradation of NRF2 via the SCF-β-TrCP complex is facilitated by phosphorylation of the Neh6 domain involving glycogen synthase kinase-3β (GSK-3β), which is recognized by β-TrCP ubiquitin ligase [22].

On the other hand, inhibitory phosphorylation of GSK-3β by extracellular signal-regulated kinase (ERK), p38 MAP kinase (MAPK), phosphoinositide 3-kinase (PI3K), protein kinase C (PKC), and protein kinase B/Akt kinase stabilizes and activates NRF2 [14]. 3-Hydroxy-3-methylglutaryl reductase degradation 1 (Hrd-1) is another ubiquitin ligase that facilitates proteasomal degradation of NRF2 [23]. The degradation complexes also can be influenced by post-translational modifications on NRF2. For example, PKC has been reported to phosphorylate NRF2 at Ser40, promoting its dissociation from KEAP1, thereby increasing NRF2 stability [24].

## 3. Epigenetic Mechanisms Regulating NRF2 Signaling

Mechanisms of NRF2 regulation discussed above can be influenced both by genetic and epigenetic alterations. Biallelic inactivation of *KEAP1,* resulting in NRF2 hyperactivation, was identified as a relatively common event in lung cancer [25]. Nuclear accumulation of NRF2 and low expression of KEAP1 correlated with tumor aggressiveness, although the expected phenotypic outcomes were not necessarily consistent in cases of somatic mutation in *KEAP1* vs. *NFE2L2* [26].

The discovery of *KEAP1* promoter hypermethylation, leading to gene silencing in lung cancer [27], prompted widespread research on epigenetic regulation of the NRF2 signaling pathway. This topic has been covered in detail in several excellent review articles [28,29,30]. Prior to discussion of the various phytochemicals reported to act via epigenetic mechanisms, a brief description of the epigenetic regulation of NRF2 signaling is presented first.

### 3.1. DNA Methylation

Several CpG islands were identified in the promoters of *NFE2L2* and *Nfe2l2* [31]. Hypermethylation of these CpG sites markedly lowered NRF2 expression in prostate tumorigenesis [32]. As discussed above, *KEAP1* promoter methylation was first reported in lung cancer, but similar observations have been linked to poor prognosis in glioma, breast cancer, non-small cell lung carcinoma, colorectal cancer, clear renal cell carcinoma, and pancreatic cancer [33,34,35,36,37,38].

### 3.2. Histone Modifications 

In addition to DNA methylation, histone methylation and acetylation, among other changes, also plays a vital role in regulating gene expression. Methylation of histones occurs primarily on the basic amino acids lysine and arginine. Gene activation or repression depends on the amino acid that is methylated and the degree of methylation, i.e., monomethylation, dimethylation, or trimethylation. Enhancer of zeste homolog 2 (EZH2), which is a member of the Polycomb group of proteins that catalyze trimethylation of histone H3 lysine 27 (H3K27me3), was downregulated in lung cancer and associated with *NFE2L2* gene silencing [39].

Another study showed that increased binding of transcription factor Specificity protein 1 (Sp1) on the *KEAP1* promoter increased methylation of histone H3K4 by the histone methyltransferase (HMT) SET Domain Containing 7 (SETD7, Set7/9) [40]; thus, *KEAP1* also is regulated by changes in histone methylation.

Histone acetyltransferases (HATs) and histone deacetylases (HDACs) also are viewed as critical epigenetic “writers” and “erasers” that catalyze, respectively, the addition and removal of acetyl groups from histone and non-histone proteins. Acetylation of NRF2 at the Neh1 DNA binding domain by the HAT p300/CREB binding protein (CBP) is required for Nrf2-dependent gene transcription [41]. Similarly, p300/CBP was implicated in the regulation of subunit p65 in NF-κB-mediated NRF2 activation, and in recruiting HDAC3 to inhibit ARE-dependent gene transcription [42]. On the other hand, HDAC2 increased the stability of NRF2 protein by deacetylating lysine residues, thereby preventing NRF2 protein degradation [43].

### 3.3. Epigenetic “Readers”

In addition to the “writers” and “erasers”, epigenetic “reader” proteins are gaining attention in the context of Nrf2 signaling. Bromodomain and extraterminal domain (BET) proteins interact with acetylated lysine residues in histones, regulating genes such as *MYC*, but also with acetylated non-histone proteins such as Nrf2 to inhibit Nrf2-dependent signaling, in species as diverse as mammals and *Drosophila* (reviewed in [44]).

Evidence linked upregulation of *KEAP1* expression with increased levels of the BET protein BRD4, and with concomitant downregulation of NRF2 in prostate cancer cell lines, where integration of RNA-seq data with chromatin immunoprecipitation (ChIP) assays correlated NRF2-dependent gene expression with *KEAP1* among the top genes interacting with BRD4 [45].

### 3.4. Regulation by Non-Coding RNAs

Non-coding RNAs linked to NRF2 signaling include miRNAs, which contain ~22 nucleotides, and lncRNAs, which are greater than 200 nucleotides in length. Other aspects of non-coding RNA biology will not be discussed here in detail, although they represent interesting avenues for future research. For example, overexpression of small nucleolar RNA *ACA11* was linked to a hyper-proliferative phenotype, reactive oxygen species generation, and NRF2 nuclear import in multiple myeloma [46]. Post-transcriptional changes to RNAs, such as 6-methyladenosine (m6A) modification of *NFE2L2* mRNA, also warrant further investigation [47]. 

The contribution of miRNAs to the regulation of NRF2 signaling is widely recognized. For example, downregulation by targeting the 3′untranslated region (3′UTR) “seed” sequences in *NFE2L2* mRNA has been reported for miR-153, miR142-5p, mi-R27a, miR-144, miR34a, and miR-93 (reviewed in [48,49]). On the other hand, an increase in NRF2 expression and activity was observed by miRNAs targeting the 3′UTR of *KEAP1* mRNA. The first miRNA reported to regulate *KEAP1* in this manner was miR-200a [50], followed by other examples, such as miR-455-3p, miR-141, miR-7, and miR-432-3p [51,52,53,54].

Several lncRNAs have been identified with regulatory roles in NRF2 activation. These lncRNAs include: (i) *UCA1*, *MEG3,* and *NRAL* acting as competing endogenous RNAs (ceRNA) for mRNA by “sponging” the respective miRNAs, (ii) *HOTAIR* increasing histone H4 acetylation at the *NFE2L2* promoter, (iii) *MALAT1* negatively regulating *KEAP1*, (iv) *TUG1* interacting directly with the NRF2 protein, and (v) *NMRAL2P* serving as a novel functional pseudogene both upstream and downstream of NRF2 (reviewed in [12]).

## 4. Phytochemicals and the Epigenetic Regulation of NRF2 Signaling

The role of dietary phytochemicals in cancer prevention and interception is well documented, and NRF2 signaling is considered a major target for many bioactive natural compounds [55,56,57]. Such agents affect the expression and activities of downstream target genes of NRF2, and the crosstalk with other signaling pathways, not only in cancer but also in other chronic diseases. However, the impact of phytochemicals on epigenetic mechanisms regulating NRF2 is an emerging area. In this section, we provide an update on the current state of knowledge regarding epigenetic regulation of NRF2 signaling by specific phytochemicals.

### 4.1. 3,3′-Diindolylmethane (DIM)

Indole-3-carbinol (I3C) is derived from the breakdown of glucobrassicin in cruciferous vegetables such as cabbage, cauliflower and broccoli [57]. Under low pH conditions in the stomach, I3C forms oligomers [58,59], including the dimer 3,3′-diindolylmethane (DIM), which have been investigated for cancer preventive and therapeutic activities [60,61,62,63,64,65,66,67,68,69,70,71,72], including in human [73,74,75,76,77,78].

In a study designed to assess epigenetic regulation of Nrf2, DIM increased *Nfe2l2* mRNA in transgenic adenocarcinoma mouse prostate (TRAMP)-C1 prostate cancer cells by reversing the methylation status of the first five CpGs in the *Nfe2l2* promoter. DIM inhibited the mRNA and protein expression of Dnmt1, Dnmt3a, and Dnmt3b, and well as Hdac2 and Hdac3 in vitro. In vivo, DIM reduced 5-methylcytosine immunostaining in prostate cancer tissues of the TRAMP mouse, which mirrors the pathogenesis of human prostate cancer, and DIM-supplemented diet lowered the incidence of palpable tumors and lymph node metastasis compared to controls [79] (Table 1).

### 4.2. Apigenin

Apigenin is a natural flavonoid derived from chamomile flowers, oranges, parsley, celery, and other natural sources, with potential antioxidant, anti-inflammatory, and anticancer properties [102]. Anticancer properties of apigenin were exhibited in many types of malignancy, and some were linked to epigenetic mechanisms [80,81,103,104]. One study showed restoration of Nrf2 expression and activity in the murine preneoplastic epidermal JB6 P+ cell line by decreasing the methylation status of the *Nfe2l2* promoter, and by inhibiting the expression of Dnmts and Hdacs [105].

Contrary to this induction of Nrf2 activity, another investigation with apigenin reported increased miR-101 levels targeting the 3′UTR of *NFE2L2* mRNA, with enhanced chemosensitivity of doxorubicin-resistant human hepatocellular carcinoma BEL-7402/ADM cells [106]. These findings point to possible species-specific differential methylation signatures for the corresponding target gene(s) in mouse epidermal vs. human hepatoma cells that have become drug-resistant.

### 4.3. Corosolic Acid

Corosolic acid is found in medical herbs, such as *Lagerstroemia speciose*, *Eriobotrta japonica*, and *Tiarella polyphylla* [107], and has gained attention for its beneficial effects in the prevention or treatment of metabolic disease, including diabetes. This triterpenoid also is reported to possess anticancer activity, and one study demonstrated effects on Nrf2 via epigenetic modifications. Specifically, corosolic acid induced *Nfe2l2* at the transcriptional level by decreasing CpG methylation in the corresponding promoter region of TRAMP-C1 cells. Increased histone H3 lysine 27 acetylation (H3K27ac) was also observed, as well as decreased trimethylation of H3K27 (H3K27Me3). Protein levels of Dnmts (Dnmt1, Dnmt3a and Dnmt3b) and Hdacs (Hdac1, Hdac2, Hdac3, Hdac4, Hdac7 and Hdac8) were inhibited in corosolic acid-treated TRAMP-C1 cells [82]. The study suggested that upregulation of Nrf2 was responsible for the inhibitory effects of corosolic acid on anchorage-independent growth of TRAMP-C1 cells, which was abrogated following Nrf2 knockdown [82].

### 4.4. Curcumin

Curcumin is the principal curcuminoid found in the rhizomes of turmeric (*Curcuma longa*), linked to anti-inflammatory, antioxidant, antitumor, and anti-diabetic effects. Curcumin is a known inducer of NRF2 and its downstream transcriptional targets, with emerging evidence for epigenetic regulation. Using bisulfite sequencing, Khor et al. [83] demonstrated that curcumin reversed the methylation status of the first five hypermethylated CpG islands in the N*fe2l2* promoter, thereby restoring epigenetically-silenced Nrf2 in TRAMP-C1 cells. Curcumin had negligible effects on DNA methyltransferases (Dnmts) at the RNA or protein level, but rather inhibited their enzymatic activity [83]. It was concluded that the epigenetic restoration of Nrf2 activity by curcumin might play a role in the prevention of prostate cancer in TRAMP-C1 mice [83].

### 4.5. Delphinidin

Delphinidin is an anthocyanidin flavonoid that contributes to the intense blue coloration in many fruits and vegetables, such as blackcurrant, eggplant, black grapes, red cabbage, and blackberries, and is abundant in pomegranate fruit extract [108,109]. Delphinidin is an anthocyanidin exhibiting potent antioxidant, anti-inflammatory, and antitumor properties. A study investigated the effects of delphinidin against skin cell neoplastic transformation by modulating the Nrf2 pathway. Delphinidin inhibited the neoplastic transformation of mouse epidermal JB6 P+ cells by 12-*O*-tetradecanoylphorbol-13-acetate (TPA). The anthocyanidin decreased the CpG methylation ratio at the *Nfe2l2* promoter resulting in upregulated mRNA and protein expression levels of Nrf2 and its target genes. The study further demonstrated downregulation of Dnmts (Dnmt1a and Dnmt3a), and class I and II Hdacs linked to reactivation of the Nrf2 pathway in JB6 P+ cells [84].

### 4.6. Fucoxanthin

Fucoxanthin is a xanthophyll carotenoid abundantly available in seaweeds. Its unique chemical structure provides a variety of biological activities, and it has been ascribed health benefits against chronic diseases such as cancer, obesity, and diabetes [110]. This carotenoid was found to activate Nrf2 signaling by causing demethylation of CpG islands in the *Nfe2l2* promoter, resulting in inhibition of TPA-induced transformation of JB6 P+ cells. Mechanistically, fucoxanthin decreased Dnmt1 mRNA and proteins levels, but was not found to alter Hdac expression levels [85].

### 4.7. Luteolin

Luteolin is a natural flavonoid present in the leaves, roots, stems, and fruits of several species of plants, such as chrysanthemum flowers, onion leaves, and celery [111]. Health benefits of luteolin have been linked to interfering with “hallmark features” of carcinogenesis such as angiogenesis, cell invasion, and metastasis. Regulation of the Nrf2 signaling pathway by luteolin is well studied and is a key mechanism through which the flavonoid is thought to exert health benefits. A few recent studies have pointed towards epigenetic modifications as a key underlying mechanism of action for luteolin. One investigation showed that luteolin inhibited proliferation of human colorectal HCT116 cells by upregulating *NFE2L2* mRNA and NRF2 protein expression. Bisulfite sequencing revealed a marked reduction in CpG methylation at the *NFE2L2* promoter, which was associated with significant reduction in expression and activities of DNMTs (DNMT1, DNMT3a, and DNMT3b) and HDACs (HDAC1, HDAC2, HDAC3, HDAC6, and HDAC7) [86]. Another study corroborated the epigenetic regulation of NRF2 by luteolin and showed reduced DNA methylation of the *NFE2L2* promoter in luteolin-treated human colon adenocarcinoma HT29 cells. In luteolin-treated cells, ChIP assays showed reduced DNMT1 binding and increased TET1 interaction at the *NFE2L2* promoter [87].

### 4.8. Pelargonidin

Pelargonidin is a plant anthocyanindin pigment producing a characteristic red-orange color in various fruits and vegetables, such as pomegranate, red radish, and strawberry. Pelargonidin is reported to possess antioxidant and anti-inflammatory properties and shows strong cytotoxicity towards various cancer cell lines. A molecular modeling approach in silico revealed that pelargonidin might inhibit the catalytic binding sites of human DNMT1 and DNMT3a [112]. In accordance with this work, pelargonidin reduced DNA methylation of the *Nfe2l2* promoter to activate Nrf2-driven gene expression in JB6 P+ cells, leading to suppression of TPA-induced neoplastic transformation. Treatment with pelargonidin decreased the expression of Dnmt1 and Dnmt3b, and Hdac1, Hdac2, Hdac3, Hdac4, and Hdac7) [88].

### 4.9. Polydatin

Polydatin is isolated from the Chinese herb *Polygonum cuspidatum,* and is a natural precursor of resveratrol, with potent anti-inflammatory properties that are beneficial against many pathologies, including atherosclerosis, neurological disorders and cancer. Polydatin was reported to induce Nrf2 by inhibiting Keap1 [113]. To elucidate the mechanisms behind the activation of Nrf2 by polydatin, an in vivo and in vitro model of non-alcoholic fatty liver disease (NAFLD) induced by high fructose was employed. Polydatin reduced fructose-induced oxidative stress and inflammation by inhibiting Keap1 and activating Nrf2. Polydatin also caused a marked increase in the levels of miR-200a, which targeted *KEAP1* to activate NRF2 signaling in response to high fructose-induced oxidative stress and inflammation in BRL-3A and HepG2 cells, and in the liver of high fructose-fed rats [89]. The study demonstrated that polydatin provided protection against fructose-induced liver inflammation and lipid deposition by activating Nrf2 and reducing oxidative stress [89].

### 4.10. Reserpine

Reserpine is an indole alkaloid, which is the principal active component found in the plant *Rauwolfia serpentina* and in other species of *Rauwolfia* sp. Reserpine is an anti-hypertensive drug and widely used to treat neurological disorders, and also possesses anticancer activity. A study showed that reserpine induced Nrf2-driven target genes to inhibit TPA-stimulated neoplastic transformation in mouse epidermal JB6 P+ cells. The relative methylation status of the CpG island at the *Nfe2l2* promoter was found to decrease with increasing concentrations of reserpine. Expression levels of Dnmt1, Dnmt3a, and Dnmt3b were decreased by reserpine treatment in JB6 P+ cells [90].

### 4.11. Resveratrol

Resveratrol is a widely investigated plant polyphenol due to its antioxidant, anti-inflammatory, and antimicrobial properties. Several studies have pointed towards a role as an epigenetic modifier, most notably in anti-aging research, with resveratrol classified mechanistically as a class III HDAC/ sirtuin-activating compound [114,115,116]. Other work in HepG2 cells treated with high glucose and in high-fat models of NAFLD found that the methylation status of the *NFE2L2* gene was increased, while that of *KEAP1* was decreased, leading to decreased NRF2 expression and activity [91]. Effects of resveratrol on the methylation status of the *Nfe2l2* promoter were shown in an earlier study involving a rat model of estrogen-induced mammary cancer [92]. Singh et al. noted “inhibition of 17β-estradiol-mediated alterations in NRF2 promoter methylation and expression of NRF2 targeting miR-93 after resveratrol treatment” as evidence for “resveratrol-mediated epigenetic regulation of NRF2 during E2-induced breast carcinogenesis” [92].

### 4.12. Sulforaphane

Sulforaphane is an isothiocyanate, abundant in the form of its precursor glucoraphanin in cruciferous vegetables such as broccoli [57,117,118] and acting via several anticancer mechanisms [93,119,120,121,122,123,124,125,126]. One major mechanism of action is through the induction of NRF2 and its downstream target antioxidant and detoxifying enzymes. In addition to modification of sulfhydryl groups in Keap1 [127], sulforaphane has been reported to act via epigenetic mechanisms [57,123,128,129,130,131,132,133], including inhibition of Dnmts and Hdacs, that also affect the methylation and acetylation status at the N*fe2l2* gene-level. This mechanism has been investigated in TRAMP C1 prostate cancer cells and in TPA-induced mouse skin JB6 P+ cells (Reviewed in [110]).

In a recent study, *Loc344887* was the most highly upregulated transcript in sulforaphane-treated human HCT116 colon cancer cells [94]. This non-coding RNA was identified as a novel functional pseudogene and renamed *NMRAL2P*, with 62% homology to the protein-coding gene *NmrA-like redox sensor 1* (*NMRAL1*). In addition to being a direct transcriptional target of NRF2, *NMRAL2P* was a downstream coactivator of NRF2-dependent *NQO1* expression in human colon cancer cells. It was further shown that *NMRAL2P* knockout HCT116 cells were less responsive to sulforaphane-induced growth inhibitory effects. This report added a further layer of epigenetic regulation to the NRF2 network [94], which also involves other non-coding RNAs altered by sulforaphane [134,135,136,137].

### 4.13. Tanshinone IIA

Tanshinone IIA is a lipid-soluble natural compound isolated from the medicinal herb *Salvia miltiorrhiza* Burge and associated with cardiovascular and cerebrovascular protective effects. The involvement of epigenetic mechanisms in the induction of *Nfe2l2* by Tanshinone IIA has been demonstrated in TPA-induced neoplastic transformation of JB6 P+ cells [95] and in in vitro and in vivo models of rifampicin-induced liver injury [96]. Hypomethylation of the N*fe2l2* promoter was mechanistically linked to potent induction of *Nfe2l2* mRNA and Nrf2 protein levels by Tanshinone IIA. While one study reported decreased expression of Dnmts and Hdac1, Hdac3, and Hdac8 by Tanshinone IIA in JB6 P+ cells [95], another found no significant changes in DNMTs, but elevated expression of DNA demethylases, especially ten-eleven translocation 2 (TET2), in human hepatocyte L02 and Hepa RG cells [96]. The latter investigation showed that Tanshinone IIA could prevent rifampicin-induced liver injury by inducing the expression of bile salt efflux pump (BSEP) and Na+/taurocholate cotransporter (NTCP), which were directly regulated by NRF2 [96].

### 4.14. Taxifolin

Taxifolin is a flavonoid found in onion, milk thistle, and in Pinaceae plants, with diverse pharmacological activities, including antioxidant, anti-inflammatory, and antimicrobial properties. Induction of Nrf2 and its downstream target genes is considered crucial for the beneficial effects of taxifolin [138]. Taxifolin was found to inhibit TPA-induced neoplastic transformation of JB6 P+ cells by epigenetically inducing Nrf2. Bisulfite sequencing showed that the flavonoid reduced the proportion of methylated CpG sites in the *Nfe2l2* promoter. At the molecular level, the protein expression levels of Dnmt1, Dnmt3a Dnmt3b, Hdac1, Hdac3, and Hdac8 were significantly decreased by taxifolin [97].

### 4.15. Ursolic Acid

Ursolic acid is a pentacyclic triterpenoid found ubiquitously in fruits, vegetables and herbs, such as cranberry, apple peel, basil, and rosemary [139]. Two studies have investigated the epigenetic regulation of Nrf2 by ursolic acid. In one investigation, the triterpenoid activated the Nrf2 pathway by demethylating the *Nfe2l2* promoter, accompanied by a reduction in the expression levels of Dnmts and Hdacs, which was linked to inhibition of TPA-induced neoplastic transformation in mouse epidermal cells [98]. In the second study, ursolic acid induced the expression of the protein methyltransferase SETD7, and knockdown of SETD7 decreased NRF2 protein and its downstream target genes in LNCaP and PC-3 cells. Also, H3K4me1 monomethylation at the *NFE2L2* promoter was reduced by SETD7 knockdown. On the other hand, treatment with ursolic acid enriched H3K4me1 at the *NFE2L2* promoter, leading to increased NRF2 signaling [99]. It was hypothesized that direct methylation of the NRF2 protein by SETD7 might be mechanistically relevant.

### 4.16. γ. Tocopherol–Rich Mixture of Tocopherols (γ-TmT)

The major forms of vitamin E comprise α-, β-, γ- and δ-tocopherols and related tocotrienols [140,141,142]. These lipophilic compounds are abundant in vegetable oils and nuts and are widely studied as agents with an impact on human health and disease pathogenesis [143,144,145,146,147]. Numerous reviews have covered the topic of vitamin E and cancer [148,149,150,151]. Recently, γ-TmT, a commercially available by-product of vegetable oil refinery containing 57% γ-tocopherol, was linked to induction of the Nrf2 pathway via hypomethylation of the *Nfe2l2* promoter in the TRAMP mouse and in TRAMP-C1 cells in vitro. This study also noted decreased protein levels of Dnmt1, Dnmt3a, and Dnmt3b in vivo compared to controls [100].

### 4.17. Z-Ligustilide

*Z*-Ligustilide is a natural benzoquinone derivative found in Chinese medicinal herbs, including *Radix Angelicae Sinensis,* and is reported to possess diverse pharmacological activities. Several studies [101,152,153,154] noted upregulation of Nrf2 and its downstream antioxidant enzymes by *Z*-Ligustilide. Su et al. [101] demonstrated induction of Nrf2 by *Radix Angelicae Sinensis* and purified *Z*-Ligustilide. Bisulfite sequencing revealed decreased methylation at the *Nfe2l2* promoter, accompanied by inhibition of Dnmts by both the plant extract and its isolated bioactive components [101].

## 5. Discussion

 In addition to genetic alterations, i.e., mutations or chromosomal rearrangements in germline and somatic cells, epigenetic mechanisms also play a crucial role in cancer development. For instance, promoter hypermethylation is associated with the silencing of many tumor suppressor genes. The preponderance of epigenetic deregulation in cancer and the reversible nature of these alterations [155,156] has focused attention on epigenetic changes as viable targets for prevention or therapeutic strategies. Epigenetic alterations represent promising targets for cancer interception across multiple stages, from early to late disease pathogenesis [156,157,158,159]. Relatively few “epigenetic” drugs have been approved as anticancer agents, including the DNMT inhibitor 5-azacytidine (Vidaza^TM^, Azadine^TM^) and the HDAC inhibitor suberoylanilide hydroxamic acid (SAHA, Vorinostat^TM^, Zolinza^TM^), with recent attention also shifting to the BET inhibitor JQ1 (clinicaltrials.gov). In general, these therapeutics are not without side-effects as monotherapies, and combination approaches with standard-of-care practices provide the most viable way forward to minimize toxicity and related concerns [157].

Many of the studies reviewed here involved analyses of CpG methylation sites in the *NFE2L2* promoter and inhibition of DNMTs and/or HDACs, without probing more deeply into other mechanistic aspects, such as the involvement of chromatin coactivator/corepressor complexes, long-range chromatin interactions, and non-coding RNAs. This leaves plenty of scope for future research. KEAP1-NRF2 signaling is a key molecular target of cancer preventive agents, including an array of phytochemicals (Table 1). These phytochemicals can induce NRF2 either by acting as Michael acceptors that interact with KEAP1 sensor thiols, or by activating phosphorylation cascades that stabilize NRF2 [55,160]. However, research also has demonstrated that dietary phytochemicals can regulate NRF2 signaling by diverse epigenetic mechanisms (Figure 1).

The best known NRF2 activator that has obtained clinical approval is dimethyl fumarate (Tecfidera), for the treatment of multiple sclerosis [161]. However, Tecfidera has several side-effects, including allergic reactions and gastrointestinal disturbance (www.PDR.net). There are a few related agents in clinical trials, such as Bardoxolone and SFX-01, a synthetic derivative of sulforaphane [161], which also exhibit less than desirable outcomes. Despite the promise from preclinical models and early clinical trials, safety, specificity, and potency issues must be resolved. All of these agents act by preventing the proteasomal degradation pathway. Given the multifactorial epigenetic regulation of NRF2, exploring other modulators that target NRF2 signaling at the transcription or post-transcription level warrants further attention (Figure 1 and Figure 2).

Among the phytochemicals reviewed here, most were shown to reverse *NFE2L2* promoter methylation by inhibiting DNMTs and attenuating certain HDACs. It is noteworthy that inhibitors directed against DNMTs and HDACs are used for the treatment of several malignancies, and the combination has been reported to produce synergistic growth inhibitory effects in cancer cells [162]. Therefore, it would be worthwhile to compare the combination of DNMT and HDAC inhibitors *vis-à-vis* Nrf2 induction by dietary phytochemicals (e.g., tea polyphenols plus sulforaphane). Moreover, a recent study showed that the combined inhibition of DNMT and HDAC activity caused de novo transcription of long-terminal repeats (LTRs) of the LTR12 family, linked to LTR-derived immunogens presented on major histocompatibility complex class I molecules [163]. Thus, phytochemical-mediated inhibition of DNMT and HDAC activities might dovetail NRF2 signaling with immunoprevention and immunotherapy.

Sulforaphane is an NRF2 inducer, as well as an inhibitor of HDAC activity and protein expression. One interesting observation is that Hdac3 expression was reduced by dietary sulforaphane in 1,2-dimethylhydrazine-treated wild-type (WT) mice, but less so in Nrf2^−/+^ mice, and that WT mice were more susceptible to carcinogen-induced colon tumorigenesis [164]. Thus, Nrf2 exerted an apparent oncogenic role in the gut, and Nrf2 status dictated Hdac inhibitory responses to sulforaphane and the extent of tumor growth suppression [164]. Another interesting avenue is the synergistic combination of sulforaphane with the BET inhibitor JQ1 in colon cancer models, targeting the non-histone protein Cell cycle and apoptosis regulator 2 (CCAR2) for acetylation and altered Wnt coactivator functionality [165]. As BET proteins are reported to interact with and inhibit NRF2 [44], the prospect of combined deacetylase and bromodomain inhibition affecting NRF2 regulation is worthy of further mechanistic investigation.

The phytochemicals reviewed here typically activate murine *Nfe2l2* epigenetically, coincident with Nrf2 induction, except in the case of apigenin. Whereas one report noted *Nfe2l2* activation via promoter hypomethylation in mouse skin epidermal cells [80], another showed miRNA-mediated inhibition of human *NFE2L2* by apigenin, leading to chemosensitization of adriamycin-resistant hepatocellular carcinoma cells [106]. It would be interesting to explore the underlying circumstances for this epigenetic regulation of NRF2 by apigenin and, for example, whether DNA methylation predominates over non-coding RNA-mediated mechanisms. Activation of NRF2 is highly regulated, both temporally and spatially, and it will be important to decipher the epigenetic mechanisms in various cell types and the respective context-specific regulation of NRF2. From the current update, the majority of studies involved a single cell type and one of two preclinical models: JB6+ mouse skin epidermal cells and TRAMP prostate tumor cells. This calls for further investigation in different cancer scenarios to better characterize the epigenetic regulation of Nrf2/Keap1 by phytochemicals.

Finally, NRF2 is a complex regulator in cancer etiology because of its yin/yang roles in prevention and promotion, dictated in part by early vs. late stages of disease pathogenesis [10,11]. Epigenetic mechanisms add an additional layer of regulation, with numerous readers, writers and erasers that interact to affect histone states and chromatin access. Epigenetic aspects of NRF2 signaling by natural phytochemicals are worthy of further investigation to better understand context-dependent mechanisms that might provide new avenues for cancer prevention and interception.

## 6. Conclusions

The use of dietary components as cancer preventive agents continues to be of great interest, and experimental evidence supports the role of nutritional factors in modulating deregulated signaling pathways during cancer initiation and progression. Our understanding of the complex mechanisms of action of dietary factors is constantly evolving. One major take-home message from the accrued literature is that no agent, dietary or otherwise, is likely to act by one mechanism alone, or to affect a single molecular target without influencing other components of a signaling network. It is increasingly documented that diet or dietary components can influence gene expression through epigenetic mechanisms, but further work is needed to truly appreciate how these mechanisms can be manipulated in a beneficial manner for disease interception in the context of NRF2 signaling.

## Figures and Tables

**Figure 1 antioxidants-09-00865-f001:**
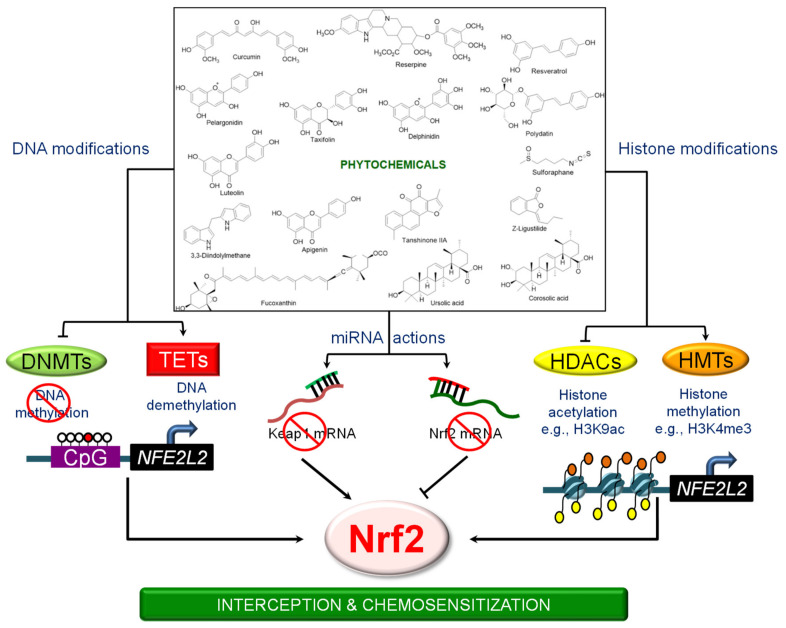
Phytochemicals and the epigenetic mechanisms linked to Nrf2-dependent signaling.

**Figure 2 antioxidants-09-00865-f002:**
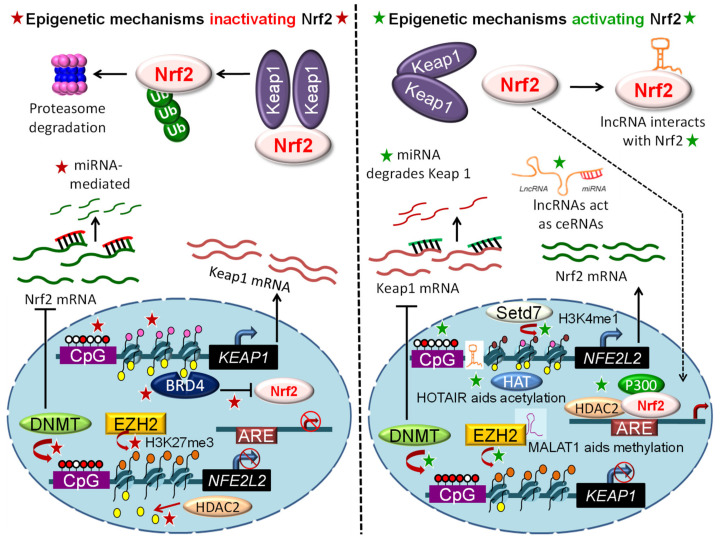
Regulation of Nrf2 signaling via epigenetically-mediated transcriptional, post-transcriptional, and post-translational mechanisms (star symbols). Red and white circles, DNA methylation and unmethylation; yellow ovals, histone acetylation; pink, brown and orange ovals, histone unmethylation, H3K4me1 and H3K27me3, respectively.

**Table 1 antioxidants-09-00865-t001:** List of phytochemicals reported to regulate NRF2 signaling epigenetically.

Sl No.	Phytochemical	Chemical Name	Epigenetic Mechanism of Nrf2 Regulation	Molecular Targets	Cell Type	Reference
1.	3,3′-diindolylmethane	3,3′-Methylenebis(1*H*-indole)	Decreased methylation of CpG sites in the promoter region of mouse *Nfe2l2*	Suppressed mRNA and protein expression of Dnmt1, Dnmt3a, and Dnmt3b; inhibited protein expression of Hdac2 and Hdac3	TRAMP-C1 prostate cells	[79]
2.	Apigenin	5,7-Dihydroxy-2-(4hydroxy-phenyl)-4*H*-chromen-4-one	Decreased *Nfe2l2* hyper-methylation; induced expression of miR-101, targeting *Nfe2l2* mRNA	Inhibited Dnmt1, Dnmt3a and Dnmt3b; inhibited Hdacs; induced miR101	Mouse epidermal JB6 P+ cells BEL-7402/ADM cells	[80,81]
3.	Corosolic acid	2α,3β-2,3-dihydroxyurs-12-en-28-oic acid	Decreased *Nfe2l2* hypermethylation; increased histone H3 lysine 27 acetylation; decreased H3 lysine 27 trimethylation	Decreased levels of Dnmt1, Dnmt3a and Dnmt3b; reduced levels of Hdac1, Hdac2, Hdac3, Hdac4, Hdac7 and Hdac8	TRAMP-C1 prostate cells	[82]
4.	Curcumin	1, 7-bis (4-hydroxy-3-methoxy-phenyl)-1, 6 heptadiene-3, 5-dione	Decreased *Nfe2l2* hypermethylation	Inhibited enzymatic activity of Dnmt enzymes	TRAMP-C1 prostate cells	[83]
5.	Delphinidin	3,5,7-Trihydroxy-2-(3,4,5-trihydroxyphenyl) chromenium	Demethylation of 15 CpG sites in the mouse *Nfe2l2* promoter region	Decreased protein expression of Dnmt1, Dnmt3a, and class I/II Hdacs	Mouse epidermal JB6 P+ cells	[84]
6.	Fucoxanthin	3,5′-Dihydroxy-8-oxo-6′,7′-didehydro-5,5′,6,6′,7,8-hexahydro-5,6-epoxy-β,β-caroten-3′-yl acetate	Decreased *Nfe2l2* hypermethylation	Reduced Dnmt activity	Mouse epidermal JB6 P+ cells	[85]
7.	Luteolin	2-(3,4dihydroxyphenyl)-5,7-dihydroxy-chromen-4-one	Decreased *NFE2L2* hypermethylation	Decreased expression of DNMT1, DNMT3A and DNMT3B; decreased HDAC1, HDAC2, HDAC3, HDAC6, HDAC7; reduced activities of DNMTs and HDACs; increased ten-eleven translocation 1, 2 and 3 (TET1, TET2, and TET3)	Human colon cancer cells and SNU-407 cells	[86,87]
8.	Pelargonidin	3,5,7-Trihydroxy-2-(4hydroxyphenyl) chromenium	Decreased methylated CpGs in *Nfe2l2* promoter	Decreased Dnmt1 and Dnmt3b expression; reduced levels of Hdacs 1–4 and Hdac7	JB6 P+ cells	[88]
9.	Polydatin	3-Hydroxy-5-[(*E*)-2-(4-hydroxyphenyl)vinyl] phenyl β-d-glucopyranoside	Enhanced miR-200a targeting *KEAP1* to activate NRF2 signaling	Increased miR-200a expression under high fructose induction; downregulated *KEAP1* mRNA and protein	Buffalo rat liver (BRL-3A) and human HepG2 cells	[89]
10.	Reserpine	Methyl 18β-hydroxy-11,17α-dimethoxy-3β,20α-yohimban-16βcarboxylate 3,4,5-trimethoxybenzoate	Decreased proportion of methylated CpG sites in the *Nfe2l2* promoter	Concentration-dependent decreased mRNA and protein expression of Dnmt1, Dnmt3a, and Dnmt3b	JB6 P+ Cell	[90]
11.	Resveratrol	3,4′,5-trihydroxystilbene	Decreased methylation of the *NFE2L2* promoter	Inhibited expression and activity of DNMT1, DNMT3a, and DNMT3b; miR93 implicated	HepG2 cells and estradiol-induced breast cancer	[91,92]
12.	Sulforaphane	1-Isothiocyanato-4-(methanesulfinyl)butane	CpG demethylation and histone acetylation at the *Nfe2l2* promoter; lncRNA upregulation	Inhibition of Dnmt1, Dnmt3a, Dnmt3b, Hdacs 1–5, and Hdac7; upregulated functional pseudogene *NMRAL2P*	JB6 P+ cells; TRAMP C1 cells; human colon cancer cells	[93,94]
13.	Tanshinone IIA	1,6,6-trimethyl-8,9- dihydro-7*H*-naphtho [1,2-g] benzofuran-10,11-dione	Decreased methylated CpGs in *Nfe2l2* promoter; increased recruitment of RNA polymerase complex II at the *NFE2L2* transcription start site	Decreased mRNA and protein levels of HDAC1, HDAC3, and HDAC8, as well as DNMT1, DNMT3a, and DNMT3b; induced expression of TET2	JB6 P+ cells, human normal hepatocyte and Hepa RG cells; rifampicin-induced liver injury in mice	[95,96]
14.	Taxifolin	(2*S*,3*S*)-2-(3,4dihydroxy-phenyl)-3,5,7-trihydroxy-2,3dihydro-4*H*-chromen-4-one	Decreased proportion of methylated CpGs in the *Nfe2l2* promoter	Reduced protein levels of Dnmt1, Dnmt3a and Dnmt3b as well as Hdacs 1, 3, 4, and 8	JB6 P+ cells	[97]
15.	Ursolic acid	(3β)-3-Hydroxyurs-12-en-28-oic acid	*Nfe2l2* mouse promoter demethylation; increased acetylation and K4 monomethylation of histone H3 in human cells	Reduced DNMT1 and DNMT3a protein levels; inhibited expression of HDACs 1-3 and 8 (Class I) and HDAC 6 and 7 (Class II); induced Setd7	JB6 P+ cells PC3 and LnCaP cells	[98,99]
16.	γ Tocopherol–rich mixture of tocopherols (γ-TmT)	(2*R*)-2,5,7,8-tetramethyl-2-[ (4*R*,8*R*)-4,8,12-trimethyl-tridecyl]-6-chromanol	Reversed hyper-methylation in the *Nfe2l2* promoter	Inhibited protein levels of Dnmt1, Dnmt3a, and Dnmt3b	Prostate tissues of C57BL/TGN TRAMP mice	[100]
17.	*Z*-Ligustilide	(3*E*)-3-butylidene-4,5-dihydro-2-benzofuran-1-one	Decreased methylation of the first five CpGs of the *Nfe2l2* promoter	Inhibited Dnmt activity	TRAMP C1 cells	[101]

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
