# Peer review of "Epigenetic Regulation of NRF2/KEAP1 by Phytochemicals"

_antioxidants, 2020, doi:10.3390/antiox9090865_

Round 1

Reviewer 1 Report

This review manuscript is properly written and well-presented about the effect of phytochemicals on Nrf2 signaling pathway. The reviewer thinks the manuscript is acceptable for Antioxidant essentially in the present style, with minor revision.

Minor
Some graphical errors are found in Figure 2 (Ub, Keap1)

Author Response

RESPONSE 1: We realized that these occurred when importing the figure into the Word template and generating the PDF –  they have been corrected in the revised iteration.

Reviewer 2 Report

This well-organized article could lead readers to a better understanding of the epigenetic regulation of the Keap1-Nrf2 signaling pathway by phytochemicals. I do recommend the study for publication after the following revisions.

Major comments:

  • Fig2: In the cytoplasm, Nrf2 is ubiquitinated by the Keap1-Cul3 ubiquitin E3 ligase complex to mark it for degradation by the proteasome. However, the schema in Figure 2 indicating ‘Inactivation of Nrf2 signaling’ could mislead the readers. The poly-ubiquitination of Nrf2 seems not to occur apart from Keap1. The diagram should be revised.
  • Although the molecular mechanism mediated with epigenetic regulation of Keap1/Nrf2 by phytochemicals is well described in Chapter 3, it is suggested that therapeutic (or not-expected) outcomes demonstrated in each study might be included in the same chapter. For instance, in addition to epigenetic modifications of Nrf2 by DIM (reversed CpG methylation status of Nrf2) as already described on page 6 (Reference No. 79), the lower incidence of tumorigenesis and metastasis shown in TRAMP mice fed with DIM-supplemented diet could be mentioned in context. These modifications would be helpful for readers to understand clinical aspects of phytochemical-use for cancer prevention and/or treatment.

Minor comments:

  • The author use Nrf2 and Keap1 to express proteins and NFE2L2and KEAP1 to express genes. However, there seem to be complications with italicization and capitalization of gene and protein names. The authors should follow the general guidelines for protein and gene nomenclature as follows:

Mice and rats:

Gene symbol: Nfe2l2or Nrf2 andKeap1

Protein symbol: Nfe2l2 or Nrf2 and Keap1

Human and non-human primates:

Gene symbol: NFE2L2 orNRF2and KEAP1

Protein symbol: NFE2L2 or NRF2 and KEAP1

  • Page 5, line 7: Abbreviation for bTrCP should be revised. There is a character corruption.
  • Please revise some items in Figure2: Parts of letters are missing (g.Ub, Keap1). ‘Proteasome degradation’ should be written with a single color for the letters.

Author Response

RESPONSE 1: Headings were revised to more precisely indicate epigenetic mechanisms activating or inactivating Nrf2, with red and green star-shaped symbols at the top to more clearly distinguish among the corresponding mechanisms shown in the figure, and to avoid confusion with other mechanisms such as ubiquitination and proteasomal turnover over Nrf2.

RESPONSE 2: As recommended by Reviewer #2, additional sentences were added in sections 4.1, 4.3, 4.4, 4.9, 4.12 and 4.13 (yellow highlighted sentences). 

RESPONSE 3: All references to protein/gene naming were corrected, as suggested.

RESPONSE 4: All relevant places in the document now refer to “β-TrCP”

RESPONSE 5: We realized that the errors in Fig 2 occurred when importing the figure into the Word template and generating the PDF –  they have been corrected in the revised iteration, and ‘proteasome degradation’ now is in black font.

Reviewer 3 Report

The manuscript is interesting and describes epigenetic regulation of Nrf2/Keap1 by phytochemicals. The aim was achieved and authors were focused to provide an update on the various phytochemicals that regulate Nrf2 via the “epigenetic trinity” of DNA methylation, histone 70modifications, and noncoding RNAs.
Unlike the genetic changes affecting DNA sequence, the reversible nature of epigenetic alterations provides an attractive 28avenue for cancer interception. Thus, targeting epigenetic mechanisms in the Nrf2/Keap1 network represents an enticing strategy for therapeutic intervention with dietary phytochemicals, acting at transcriptional, post-transcriptional and post-translational levels.
I agree with the authors that this opens an avenue for exploring additional dietary phytochemicals that regulate the epigenome, and the potential for novel strategies to target the Nrf2/Keap1 signaling axis with a view to beneficial interception of cancer and other chronic diseases.

Author Response

We thank the reviewer for positive feedback.